# FicClaim: A Framework for Claim Verification in Fictional Domains Using Synthetic Data Generation

## Abstract

The spread of misinformation and disinformation on social medial platforms has made automatic claim verification an important concern in various domains. We study the problem of claim verification in the context of claims about fictional stories for the purpose of uncovering logical inconsistencies also known as plot holes. To this end, we first introduce FicClaim, a synthetic dataset containing plot holes. FicClaim is generated in part by large language models (LLMs) for learning how to apply claim verification to fictional settings. We then develop the FicVer algorithm for finding inconsistencies in a story based on our dataset. We benchmark our algorithm against various claim verification methods and demonstrate that the proposed algorithm leads to state-of-the-art performance. Our code is available at `https://anonymized`.

## 1 Introduction

Proliferation of misinformation and disinformation on social media platforms has led to a large body of research on *claim verification* (Oshikawa et al., 2020), where the goal is to determine the truthfulness or accuracy of a given claim or statement using computational methods. Claim verification usually involves analyzing textual information and comparing and contrasting it against evidence or knowledge extracted from a large collection of factual documents, often sourced from trusted repositories such as Wikipedia or specialized academic corpora, to assess the validity of the claim. Claim verification is commonly used in applications such as news verification (Ibrishimova & Li, 2020), rumor detection (Hamidian & Diab, 2019), and detecting false or misleading information on social media (Barrón-Cedeno et al., 2020). Despite these advances, there remain numerous practical challenges beyond the realm of these applications that could greatly benefit from the application of automated claim verification methods. Particularly, these challenges pertain to scenarios where a predefined set of factual information, distinct from the specific work in question, is not readily accessible, such as in the domain of fictional literature and works.

For example, people constantly create and consume massive amounts of entertainment content through novels, TV shows, and movies. Oftentimes, the creation of these works involves substantial financial commitments, and their success amongst critics decides whether or not that investment pays off. An essential element of these works lies in their capacity to engross and captivate readers or viewers through storytelling, and one obstacle that can hinder that experience is the presence of logical disconnects, commonly known as "plot holes". Plot holes have the potential to confuse audiences and create the perception of poor quality. Consequently, the accurate identification of plot holes emerges as a practical and significant problem worth addressing. In this work, we aim to formulate the problem of plot hole detection within the context of claim verification and develop an algorithm to address this challenge.

Although plot hole detection shares similarities with fake news detection, it fundamentally differs from regular claim verification tasks in that it generally occurs in a "new" setting: it concerns itself with "facts" in a fictional world, where a set of ground truth evidence statements is not well-defined. This property means that regular state-of-the-art claim verification techniques cannot be easily applied without assuming a set of common-sense facts and risking a high rate of false positive faulty claims because these facts may not be fictionally true. For instance, in the world of *Harry Potter*, magic clearly exists. However, this fictional existence directly contradicts the laws of physics and common sense of reality. As a result, assuming common-sense facts via the use of general-purpose evidence-based methods would inevitably flag most of the work as a "plot hole". Even if we were to use the extensive fan material and fan-made wiki pages available for *Harry Potter* specifically, the approach would still be of very little use to us for

any other new (or even unreleased) works that would benefit the most from automatic claim verification methods. As a result, existing works likely will not have optimal performance for identifying plot holes.

We develop a framework for claim verification in fictional domains. We first introduce FicClaim, a synthetically-generated dataset for this problem via leveraging large language models (LLMs), and then use it to define a "plot hole problem" that we then address. We intend for FicClaim to be used as a benchmark dataset for developing models that are to be used for application in the entertainment industry on fictional works. The algorithm we develop to address this problem, FicVer, utilizes sentence embeddings in conjunction with knowledge graphs (KGs) and graph neural networks (GNNs) (Hu et al., 2021; Kipf & Welling, 2016) to extract contextual information from each document and then make predictions. By applying our method on FicClaim and comparing it to a variety of models from prior claim verification works, we show that current methods are still insufficient to reach human annotator level, and therefore practical adoption in the industry. We hope our work serves as a starting point for future works on the application of claim verification models and tools in the entertainment industry to automate tasks such as plot hole scanning. Our results show that our pipeline can potentially outperform humans annotators with statistical significance.

## 2 Related Works

Our novelty relies on adopting claim verification tools for real-world domains to work well for detecting plot holes.

**Claim Verification.** Current work on claim verification has focused mainly on tagging disinformation in social media spheres. As a result, the primary emphasis has been on fact-checking claims by leveraging knowledge repositories like Wikipedia (Thorne et al., 2018) or news sources such as Politifact (Vlachos & Riedel, 2014) as ground truth. Hence, common sense knowledge repositories are essential for fact checking for most exiting approaches because external information sources in the claim verification landscape are necessary. The most effective techniques for solving these problems thus far generally follow a three-step pipeline which was first introduced along with the FEVER dataset (Thorne et al., 2018). These three steps are document retrieval, sentence-level evidence retrieval, and textual entailment which gauges the logical relationship between the claim and the extracted evidence.

The majority of recent work has focused on the textual entailment step, after retrieving a number of sentences as evidence (Bekoulis et al., 2021). In terms of solution approach, current claim verification methods can be split into two, broad categories of pattern-based methods, e.g., LSTM (Hochreiter & Schmidhuber, 1997) and TextCNN (Wang, 2017), and evidence-based methods, e.g. DeClarE (Debunking Claims with Interpretable Evidence) and HAN (Hierarchical Attention Networks) (Popat et al., 2018; Ma et al., 2019). While some evidence-based methods such as GET (Graph-based sEmantic sTructure) use graph representations and GNNs already (Bekoulis et al., 2021; Xu et al., 2022), we propose to improve upon these works in two aspects. First, we use a deeper semantic knowledge graph as opposed to one reflecting word proximity. Second, we generate graphs for the same document on which the fact-checking occurs, thereby rendering our algorithm a "context-based" method which is more suitable for fictional domains as opposed to an "evidence-based" method that relies on common sense repositories.

**Graph Structured Representation for Claim Verification.** The utilization of knowledge graphs (KGs) has become an effective approach to address claim verification because KGs can encode relationships efficiently, which in turn provides a systematic representation for effective reasoning. It has been demonstrated that claim verification can be successfully carried out by employing classical, search-based reasoning techniques on KGs (Shiralkar et al., 2017). Consequently, considerable efforts in automatic claim verification have been devoted to exploring strategies for extracting knowledge triples and factual information from unstructured documents (Ye et al., 2021), as well as organizing this extracted data into KG structures (Tchechmedjiev et al., 2019).

On the other hand, the integration of graph neural networks (GNNs) with KGs has been identified as a promising approach for enhancing the accuracy and efficiency of performing claim verification because GNNs naturally encode relationships to perform a prediction task (Zhu et al., 2021). Prior work has also shown the efficacy of GNNs in modelling data (Schlichtkrull et al., 2018) and for making decisions based on surrounding context, albeit outside of NLP (Chadalapaka et al., 2023). Building upon these works, we explore how knowledge representations provided by KGs can be leveraged in the context of deep learning models applied to decision making in fictional domains where pre-established facts are unavailable. Specifically, we apply a similar approach to constructing KGs as previous works on building narrative chains (Chambers & Jurafsky, 2008), with the exception that we allow our KGs to contain loops.

By examining this aspect, we aim to demonstrate that KG-based knowledge representations can effectively support using GNNs even in domains that lack concrete factual foundations. To this end, we also offer a new domain-specific dataset that be used for benchmarking in follow-up research.

## 3 Problem Description

A key characteristic of claim verification in the context of fictional domains is that it would require out-of-domain prediction relative to most existing training datasets. As a result, this problem requires a solution with low-shot capability that can easily be used as a proof-of-concept to demonstrate our framework. To be specific, we focus on the detection of a certain class of plot hole known as a *continuity error*. A "continuity error" means that a logical inconsistency is found in the story. More formally, we define a document $X$ made up of $n$ sentences (aka claims) $X = \{X_1, ..., X_n\}$, with $n = |X|$. A "continuity error" occurs if there exists some sentence $X_i$ such that the remaining sentences entail the opposite meaning of $X_i$, i.e. $X \setminus X_i \models \neg X_i$. For our purposes, given a story with an error, we consider a solution to the problem to be the index $i$ (or indices, in the case of multiple plot holes) of the problematic sentence(s) $X_i$ in the document that entails a plot hole. Based on this proof-of-concept problem definition, we present a framework for addressing similar problems and effectively leveraging available information in fictional domains. At a high level formulation, our contributions consist of the following key steps:

1. We introduce FicClaim, a synthetic dataset specifically tailored to the characteristics of our problem. This initial step is crucial as it allows us to evaluate and test our proposed algorithms effectively. We generate the dataset synthetically because finding a real-world dataset or manually generating it for our problem is challenging. Additionally, by generating a synthetic dataset, we can ensure that we have control over the data distribution, the amount of introduced inconsistency, and the specific scenarios we want to explore.

2. We propose a solution that revolves around constructing a KG and then training a GNN-based model to tackle the downstream predictive task. The KG serves as a structured representation of the relationships and connections within the fictional domain, enabling the model to learn and reason about the data effectively. By leveraging GNNs, which excel in capturing and encoding complex dependencies in graph-structured data, we can effectively extract valuable insights and make accurate predictions using a relatively small dataset.

## 4 FicClaim Dataset

To the best our knowledge, no prior work has explored the problem of claim verification in fictional domains. Thus, the first obstacle that we face is that current datasets that use real-world external knowledge are not suitable for evaluating our scenario. For that reason, we develop a benchmark for testing the algorithms we propose. To that end, we apply a novel synthetic generation process via the use of LLMs on 1,000 fictional stories with at least 200 words pulled from the r/WritingPrompts and r/stories subreddits and sourced from Kaggle [1], resulting in 30,000 synthetic datapoints, split into three datasets as described below. Within each dataset, 8000 samples were used for training, and the other 2000 samples were used for testing. The core idea is to introduce plot holes in the dataset such that they look semantically natural.

Since we are primarily interested in scenarios which lack labelled datasets, we benefit from synthetic data generation to create a test-bed for our algorithm. To generate a synthetic dataset, we deliberately inject plots holes (i.e., false claims) into the documents in the place of real claims, which we then use as our labels. Furthermore, in order to limit the number of pre-existing plot holes in the dataset and therefore increase the accuracy of our synthetic labels, we ensure that all of the original stories have at least 1000 upvotes. Although this method is not perfect, our approach is similar to the way in which the FEVER dataset (Thorne et al., 2018) is generated and will provide a reasonable dataset for evaluating algorithms.

Specifically, we employ a generation approach based on negating sentences in the dataset. For a given document $X$, we define the semantic negation of a sentence $X_i$ to be some claim $X_i'$ such that $X_i \wedge X_i'$ is always false. To generate a semantic negation, we then ask the *gpt-3.5-turbo-instruct* model via an OpenAI API (OpenAI, 2023) to negate a sentence using the prompt, "Negate the following: $X_i$", the output of which is denoted $InverseGPT(X_i)$.

---

[1] https://www.kaggle.com/datasets/trevordu/reddit-short-stories

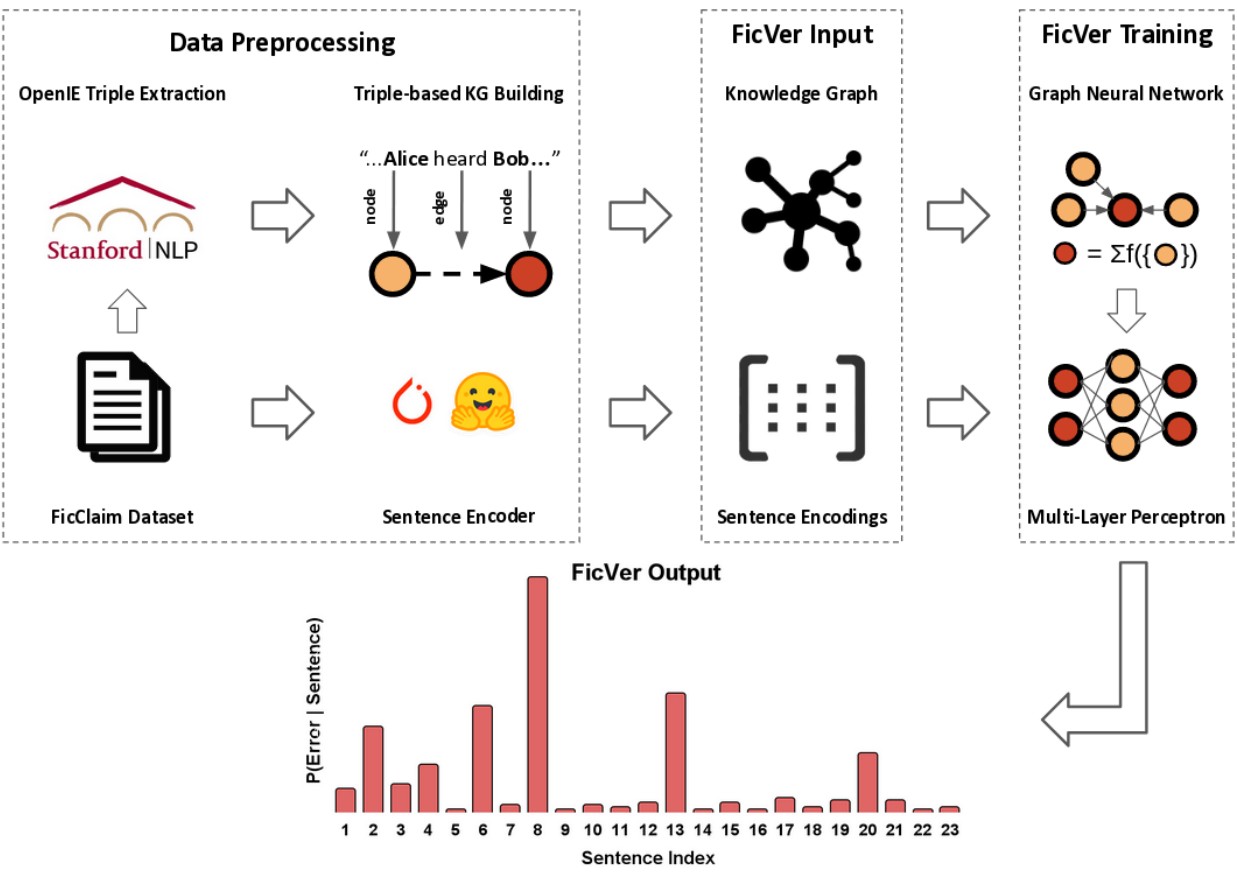

Figure 1: A block-diagram visualization of our proposed pipeline: First, data processing is done on the FicClaim dataset to create sentence encodings and a knowledge graph for each story in the dataset. Then, a GNN-based model is trained on both the encodings and the knowledge graphs to perform the predictive task of plot hole prediction.

Alternative sentence negation strategies that we tried before using LLMs are discussed in the appendix, in Sec. A.1. Examples of our negation method are provided in the appendix, in Sec. A.2.

We then inject a series of $m$ continuity error into a story $X$ by repeatedly and uniformly choosing a random sentence $X_y$ from it and replacing it with its semantic negation $InverseGPT(X_y)$, as presented in Algorithm 1. Each sentence is then labelled as either 0 or 1, with 1 meaning that the sentence has been negated. This synthetic data generation is repeated for $m = 1, 2, 5$ to create three datasets with 1, 2, and 5 plot holes per story respectively. For simplicity, negative examples (i.e. unaltered stories) are not included in our final dataset. The resulting three datasets could additionally be blended to produce a larger dataset with a varying number of plot holes, but for this work, we choose to keep the datasets separate and focus on the simpler problem of finding a fixed number of plot holes. Further discussion on the real-world application of this dataset is in the appendix, in Sec. A.3.

Based on empirical exploration with human subjects, we have determined that humans are anticipated to achieve an approximate $F1$ score of 0.5 on this dataset for a single continuity error. We have included our dataset as a supplementary resource for future explorations. We utilize this dataset, which we refer to as FicClaim, to evaluate our algorithm, which we refer to as FicVer.

## 5 Proposed Method

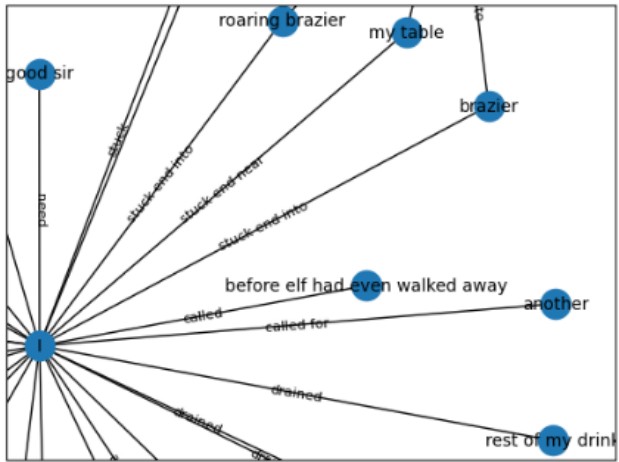

Figure 2: Example KG constructed from using all Stanford CoreNLP OpenIE triples.

Figure 1 present the block-diagram level visualization of our data processing and prediction pipeline. Our method first performs a data preprocessing step to generate KGs and sentence encodings from the dataset to make it structured. We then train a GNN-based model, FicVer, on the resulting processed data, to solve the predictive task.

### 5.1 Dataset Preprocessing

Each of the stories as a data point is preprocessed twice. First, we generate an encoding of the story to convert it to an embedding via sentence-by-sentence processing. We then a generate a knowledge graph (KG) of the story which is used in our predictive GNN model.

**Story Encoding.** Each story in the dataset is first encoded sentence-by-sentence using a BERT model (Devlin et al., 2018) fine-tuned for producing high-fidelity 384-dimensional semantic sentence encodings called a SentenceTransformer (Reimers & Gurevych, 2019). The process results in an encoding of size $(n, 384)$, with $n$ being the number of sentences in the given story. Since different stories have a different number of sentences in

---

**Algorithm 1** Continuity Error Injection Method for Generating the FicClaim Dataset

**Require:** document $X$, numErrors $m$
  $M \leftarrow \emptyset$
  **while** $|M| < m$ **do**
    $y \leftarrow unif\{1, len(X)\}$
    **while** $y \in M$ **do**
      $y \leftarrow unif\{1, len(X)\}$
    **end while**
    $M \leftarrow M \cup y$
    $X'_y \leftarrow InverseGPT(X_y)$
    $X_y \leftarrow X'_y$
  **end while**
  **return** $X, M$

---

general, all stories are then zero-padded to meet the length of the longest story for consistency. This process results in a final story encoding of size $(N, 384)$ for all stories, where $N$ is the length of the longest story in the entire dataset.

**Knowledge Graphs Construction.** Since we can't use pre-established facts to determine claim verification for our tasks, KGs can be used to formalize the contextual information from within each story that we can then actually use for verification. To encode entities and their relationships in a story, we use the Stanford CoreNLP library (Manning et al., 2014) after tokenization and lemmatization to extract KGs from a story through the following four steps:

1. Named entity recognition, to extract what the nodes should be in the generated KG.

2. Coreference resolution, which helps build more meaningful and accurate KGs.

3. Relation extraction (or triple extraction), in which knowledge triples (or subject-relation-object triples) are extracted on a sentence-by-sentence basis. From this point onwards, subjects and objects are considered "nodes" while the relations are considered to be "edges".

4. Node and edge embedding, in which nodes are one-hot encoded to represent which entity they are representative of, and a semantic encoding of relation text for edges is extracted via Word2Vec.

The specific construction of our KG is determined by the knowledge triples that we receive from the Stanford CoreNLP OpenIE triple extraction system. For many sentences, the CoreNLP program returns a set of knowledge triples that may overlap, but that contain the same information, e.g., "Alice went to see Bob" may return all of the knowledge triples (Alice, went to see, Bob), (Alice, saw, Bob), (Alice, went, Bob). For our work, we decided to include *all* of these overlapping triples to ensure that our KGs were as connected as possible, since in some cases, the subject and object of the triple are slightly different across different sentences, and we require a perfect text match in order to consider any subject or object as being the same node in the KG. As a result, although our KGs are noisy, we find that they are sufficiently connected for our algorithm to process them. Possible future work on improvements to this process is discussed in Sec. 8.

An example KG outputted from our proposed flow is shown in Fig. 2, where the sentences represented in the KG (along with some accompanying triples as shown in the example KG) are:

1. "I stuck the end of my cigarette into the roaring brazier near my table.": (I, stuck end near, my table), (I, stuck end into, roaring brazier), (I, stuck end into, brazier)

2. "My patron sipped at his booze like a bird and grimaced, while I drained mine and called for another before the elf had even walked away.": (I, called, before elf had even walked away), (I, called for, another), (I, drained, rest of my drink)

3. "I need your help, good sir.": (I, need, good sir)

The end result of this process is a set of three variables which comprises a single KG: node embeddings of shape $(n_{nodes}, d_n)$, edges, and edge embeddings of shape $(n_{edges}, d_e)$, where $n_{nodes}$ and $n_{edges}$ are the number of nodes and edges in the generated graph respectively, $d_n$ is the maximum number of entities in any given story, and $d_e$ is the Word2Vec embedding size. On average, KGs generated using this method with this dataset have 99.1 nodes and 107.8 edges, with a little over 15 connected sub-graphs per KG in total. We then use a GNN to process them as their inputs. Since these KGs are also the target of our ablation study, our model can also be configured to ignore them.

## 5.2 FicVer Architecture and Algorithm

FicVer uses a vanilla transformer model (Vaswani et al., 2017) which uses sentence-wise embeddings in conjunction with GNNs to process the KG input to improve the efficacy of our pipeline. The transformer and the GNNs are optimized simultaneously. Unlike many other claim verification models, individual tokens are not considered.

### 5.2.1 Transformers

The transformer portion of our model takes a sequence of encoded sentences as input, and processes them by first taking them to an embedding dimension and through a series of transformer encoder layers. From there, a set of probabilities over all of the sentences which represent each sentence's likelihood of containing a continuity error, i.e., $P(Error|Sentence)$, are computed. This computation is done via a standard multi-layer perceptron (MLP), followed by a softmax layer. Since KGs are being utilized, the output of the GNN processing the KG is concatenated to each sentence encoding before applying the downstream MLP. As a result, entity relations are encoded through GNNs and temporal information through the transformer which are merged to compute $P(Error|Sentence)$.

### 5.2.2 Graph Neural Networks (GNNs)

To make use of the KGs generated from our processing pipeline on FicClaim, we use GNNs. GNNs are a good choice for our problem because they are capable of benefiting from and modeling the contextual information in the input story graph. This process is done via iterative, neighborhood-level aggregations (called *graph convolutions*) at each node on the graph to update node features based on the neighboring nodes. Formally, given a story graph $G = (V, E)$, where $V$ is the set of vertices and $E$ is the set of edges, let $h_u^l$ be the $d$-dimensional node features for a node $u \in V$ at

a layer $l$ in the GNN and $h_{uv}$ be the $d'$-dimensional edge features corresponding to the edge between nodes $u$ and $v$. Let $N(v)$ be the set of nodes neighboring $v$. Then, a GNN operates on the data as follows:

$$\begin{aligned}
\mathbf{h}_{uv}^l &= \rho(\mathbf{h}_u^l, \mathbf{h}_v^l, \mathbf{h}_{uv}; \Theta_\rho^l) \\
\hat{\mathbf{h}}_u^l &= \zeta(\{\mathbf{h}_{uv}^l | v \in N(u)\}; \Theta_\zeta^l) \\
\mathbf{h}_u^{l+1} &= \phi(\hat{\mathbf{h}}_u^l, \mathbf{h}_u^l; \Theta_\phi^l)
\end{aligned} \tag{1}$$

where $\rho(\cdot)$, $\zeta(\cdot)$, and $\phi(\cdot)$ represent functions with learnable parameters whose definitions differ depending on the type of GNN used (Bronstein et al., 2017). Since the specific GNN that we choose can affect the results, we choose to test two different types of GNNs to process our KGs. First, we test a more expressive variant of the popular graph attention network (GAT) (Velickovic et al., 2017) called the graph attention network v2 (GATv2) (Brody et al., 2022). GATv2 convolutions can be described by the following $\rho$, $\zeta$, and $\phi$:

$$\begin{aligned}
\rho &= \mathbf{a}^{l^T} LeakyReLU(\mathbf{W}^l \cdot [\mathbf{h}_u^l || \mathbf{h}_v^l]) \\
\zeta &= \sigma(\sum_{v \in N(u)} softmax_v(\mathbf{h}_{uv}^l) \cdot \mathbf{W}^l \mathbf{h}_j) \\
\phi &= \hat{\mathbf{h}}_u^l
\end{aligned} \tag{2}$$

where $\Theta_\rho^l = \{\mathbf{a}^l, \mathbf{W}^l\}$, $\Theta_\zeta^l = \{\mathbf{W}^l\}$ (the same weights as in $\Theta_\rho^l$), and $\Theta_\phi^l = \emptyset$, for $\mathbf{W}^l \in \mathbb{R}^d$ and $\mathbf{a} \in \mathbb{R}^{2d}$. Second, we use a standard graph convolutional network (GCN) (Gilmer et al., 2017), which simply sums neighboring nodes and applies a $ReLU$ nonlinearity after performing a linear projection. The operation is described as:

$$\begin{aligned}
\rho &= \mathbf{h}_v^l \\
\zeta &= ReLU(\sum_{v \in N(v)} \mathbf{W}^l \mathbf{h}_{uv}^l) \\
\phi &= \hat{\mathbf{h}}_u^l
\end{aligned} \tag{3}$$

where $\Theta_\rho^l = \emptyset$, $\Theta_\zeta^l = \{\mathbf{W}^l\}$, and $\Theta_\phi^l = \emptyset$, for $\mathbf{W}^l \in \mathbb{R}^d$. Finally, we test a version of FicVer which ignores the KG and uses no GNNs as an ablation study to ensure that the use of GNNs and computation of KGs is effective and leads to an optimal performance.

# 6 Experimental Setup

Our implemented code is provided as a supplement. We first offer our experimentation and evaluation setup.

## 6.1 Baselines for Comparison

We provide three classes of claim verification methods for comparison against FicVer: pattern-based methods, evidence-based methods, and also "sanity check" methods which we use to estimate the higher-end and lower-end of estimated model performance via human annotators and random noise respectively. Our goal in providing all of these methods is to provide a comprehensive evaluation that we can use to demonstrate that our method is competitive and to examine the shortcomings of exiting methods in the context of fictional domains. We additionally experimented with an LLM baseline, but did not include it in our final results due to budgetary constraints discussed in the appendix, in Sec. A.4. Regardless, a working proof-of-concept for applying LLMs is provided with our code.

### 6.1.1 Sanity-Check Baselines

**Human Annotators.** For the human annotator baseline, we manually searched for plot holes in a small subset of documents. Due to the labor-intensive nature of this method, this was only performed on the one error dataset. This baseline serves as a higher-bound of performance expectations.

**Noise.** The random noise baseline uses noise to randomly decide if a sentence is a plot hole or not. This baseline serves as a lower-bound of performance expectations. Improvements over it demonstrate the strength of a method.

### 6.1.2 Pattern-based methods

**LSTM (Huang et al., 2015).** Sentence-wise embeddings are sequentially fed into a bidirectional LSTM. The model then takes all hidden states to a single value embedding and uses a softmax function to find the probability of error for each sentence.

**TextCNN (Wang, 2017).** A series of convolutional layers are applied to each sentence in a story at the level of tokens to determine if the sentence is an error.

### 6.1.3 Evidence-based methods

Unlike the previously-described methods, evidence-based methods decide truthfulness on a claim-by-claim basis as opposed to a document-by-document basis. For this work, the "evidence" passed was substituted with the entirety of the original text instead (i.e., the context). The "claim" was then substituted with each of the sentences in the story as a separate data point, in such a way that each document in the training set corresponded to approximately 100-200 samples batched together as opposed to a single sample, since each story had approximately 200 different sentences.

**DeClarE** (Popat et al., 2018). BiLSTMs followed by attention mechanisms are applied to the set of sentences in the story, which allows comparison between various sentences in a story.

**GET** (Xu et al., 2022). Both word- and sentence-level attention embeddings are computed, with the addition of graphs that are built using word proximity. GNNs are then employed to process the graphs, which then represent connections between the given claims.

## 6.2 Evaluation Metrics

FicClaim is scored using $F1$ score, as is standard for regular claim verification tasks in the literature. A higher $F1$ score is considered better. Additionally, both macro and micro scores are provided for the $F1$, precision, and recall. The best score is taken for a model over 5 separate runs done with different random seeds, and statistical significance was computed via Welch's $t$-test. Results were considered to be significant if $p < 0.05$. For additional context, we also provide the micro and macro scores for $F1$, precision, and recall due to the large class imbalance.

## 6.3 Ablation Study

We test all models in three different plot hole detection scenarios: 1, 2, and 5 plot holes (Tab. 1). Varying the number of plot holes in this manner allows us to determine the performance of each model with different problem configurations that match the real world, where it is possible for more than one mistake to be made within a single document.

The results of our ablation study can also be found in the same table for the different configurations of FicVer using a GATv2 vs the GCN vs using no GNNs at all. These comparisons allow us to understand the effects of using attention to process KGs, and also the efficacy of using KGs in this scheme in the first place.

## 6.4 Implementation Details

All models were trained for 200 epochs with a 80-20 train-test split, except for evidence-based models which were each trained for 20 epochs. This choice was to offset the fact that, as discussed, both evidence-based models are trained on a claim-by-claim basis as opposed to a story-by-story basis, combined with the fact that there are approximately 100x more claims than there are stories (since each story has roughly 100 sentences on average). This means that the total dataset size, training time, and number of times that each evidence-based model sees each document were all effectively 100 times greater per epoch than with the pattern-based models, warranting some level of normalization. Hyperparameters were unchanged from the original implementations of each model. A precision-recall threshold of

| # Plot Holes | Model | F1-T | P-Ma | P-Mi | R-Ma | R-Mi | F1-Ma | F1-Mi |
|---|---|---|---|---|---|---|---|---|
| | Noise | 0.008 | - | - | - | - | - | - |
| | Human* | 0.500 | - | - | - | - | - | - |
| | LSTM | $0.033 \pm 0.000$ | 0.508 | 0.827 | 0.831 | 0.827 | 0.469 | 0.827 |
| | TextCNN | $0.040 \pm 0.027$ | 0.509 | 0.947 | 0.633 | 0.947 | 0.506 | 0.947 |
| 1 | GET | $0.000 \pm 0.000$ | 0.497 | 0.995 | 0.500 | 0.995 | 0.498 | 0.995 |
| | DeClarE | $0.017 \pm 0.000$ | 0.502 | 0.300 | 0.623 | 0.300 | 0.234 | 0.300 |
| | FicVer | $0.550 \pm 0.005$ | 0.766 | 0.996 | 0.774 | 0.996 | 0.770 | 0.996 |
| | FicVer+GCN | $0.575 \pm 0.019$ | 0.784 | 0.996 | 0.789 | 0.996 | 0.786 | 0.996 |
| | FicVer+GATv2 | $\mathbf{0.586 \pm 0.023}$ | 0.788 | 0.997 | 0.797 | 0.997 | 0.792 | 0.997 |
| | Noise | 0.017 | - | - | - | - | - | - |
| | LSTM | $0.065 \pm 0.000$ | 0.516 | 0.797 | 0.895 | 0.797 | 0.476 | 0.797 |
| | TextCNN | $0.075 \pm 0.055$ | 0.518 | 0.902 | 0.732 | 0.902 | 0.512 | 0.902 |
| 2 | GET | $0.000 \pm 0.000$ | 0.495 | 0.991 | 0.500 | 0.991 | 0.497 | 0.991 |
| | DeClarE | $0.021 \pm 0.003$ | 0.504 | 0.414 | 0.661 | 0.414 | 0.301 | 0.414 |
| | FicVer | $0.545 \pm 0.023$ | 0.851 | 0.994 | 0.721 | 0.994 | 0.771 | 0.994 |
| | FicVer+GCN | $\mathbf{0.551 \pm 0.015}$ | 0.874 | 0.994 | 0.716 | 0.994 | 0.774 | 0.994 |
| | FicVer+GATv2 | $0.544 \pm 0.011$ | 0.863 | 0.994 | 0.716 | 0.994 | 0.770 | 0.994 |
| | Noise | 0.041 | - | - | - | - | - | - |
| | LSTM | $0.159 \pm 0.003$ | 0.541 | 0.859 | 0.804 | 0.859 | 0.541 | 0.859 |
| | TextCNN | $0.169 \pm 0.133$ | 0.545 | 0.875 | 0.799 | 0.875 | 0.551 | 0.875 |
| 5 | GET | $0.000 \pm 0.000$ | 0.489 | 0.978 | 0.500 | 0.978 | 0.494 | 0.978 |
| | DeClarE | $0.058 \pm 0.015$ | 0.513 | 0.462 | 0.695 | 0.462 | 0.340 | 0.462 |
| | FicVer | $0.402 \pm 0.019$ | 0.945 | 0.986 | 0.629 | 0.986 | 0.697 | 0.986 |
| | FicVer+GCN | $\mathbf{0.415 \pm 0.015}$ | 0.928 | 0.986 | 0.636 | 0.986 | 0.704 | 0.986 |
| | FicVer+GATv2 | $0.409 \pm 0.013$ | 0.934 | 0.986 | 0.632 | 0.986 | 0.701 | 0.986 |

Table 1: Model performance comparison results. Bolded values indicate best F1 scores for each section. $P$, $R$, and $F1$ refer to the precision, recall and F1 scores respectively, and -Ma and -Mi refer to the macro and micro scores. F1-T refers to the binary F1 score. All binary F1 scores are accompanied with a 95% confidence interval. - indicates unavailable data. * indicates that the human benchmark is only an approximation, given that humans annotators were only shown a smaller randomized subset of the full dataset.

0.3, chosen empirically, was used for all models to decide whether or not $P(Error|Sentence)$ would be interpreted as a plot hole or not. Further negation details are described in our appendix, in Sec.A.4.

# 7 Results

## 7.1 Comparative Results

Table 1 shows the comparison between FicVer and alternate claim verification methods on the FicClaim dataset. Additionally, Figure 3 provides a visual representation of the results. A close observation of these results indicates that existing methods do not generalize well to claim verification in fictional domains. We thus conclude that our problem is a well-defined research problem which deserves further investigation.

The best-scoring model was our FicVer+GATv2 model, which obtained $F1$ scores that were significantly better than all of the other models. We believe that the reason for the improved performance is twofold: first, looking at sentence-level semantic embeddings instead of word-level embeddings makes it intuitively simpler to search for sentences that are semantically opposed or contradicting (e.g., via an inner product), which could also have made it easier to look for logically conflicting sentences as well. Second, by including knowledge triples in the inference process through the use of GNNs and KGs, it is possible that the model could have gleaned a deeper understanding of the context behind specific sentences, therefore giving it more information about which claims are "opposing". Both of these hypotheses are supported by closely analyzing the outputs of the FicVer+GATv2 model in the single-error case, shown in Figure 4. Specifically, we find that examining the cases where the FicVer is not completely certain in its choice,

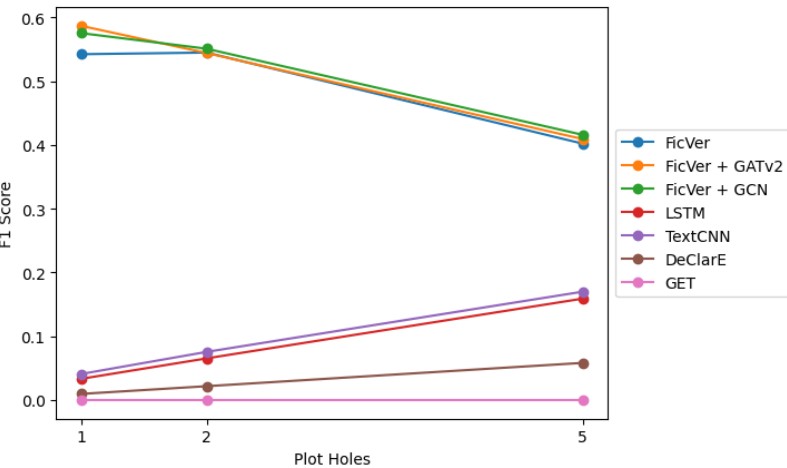

Figure 3: Relationship between the number of plot holes and $F1$ score. $F1$ score tends to increase as the number of inconsistencies increases for lower-performing models, but tends to decrease as the number of plot holes increases for better-performing models.

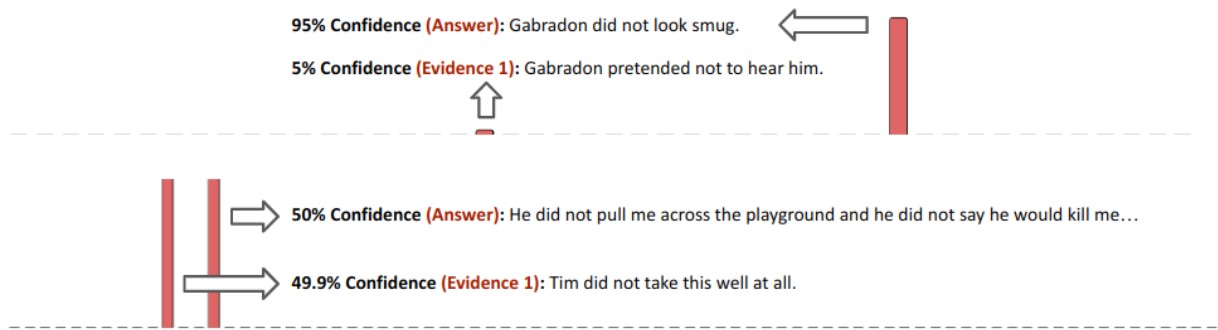

Figure 4: Analysis of FicVer+GATv2 outputs. The outputs show that in some scenarios, FicVer's next-best choices could potentially be interpreted as evidence.

e.g. $P(Error|Sentence) <= 0.95$, can shed light on the decision-making process learned by the model. Figure 4 shows two such cases that we found in the training dataset. Taking the first example from the figure, we see that the next-best sentence reveals what could be interpreted as the "opposing" claim that we expected. The solution sentence, "Gabradon did not look smug," could reasonably be refuted by the statement "Gabradon pretended not to hear him," given that aloofness is a potential indicator for smugness. The second example showcases something similar, but the nearly even split in likelihoods assigned by the model suggests that there are scenarios in which evidence may not point to one definitive plot hole if there are not enough relevant evidential claims made in the rest of the story. The hypothesis regarding the processing of knowledge triples via GNNs are supported by our ablative results (Table 1), which show that the additional use of GNNs results in higher overall performance. Our results show that, in general, using a GCN leads to the best performance for processing KGs with FicClaim.

Next best performances were achieved by the pattern-based class of models LSTM and TextCNN, both of which performed nearly identically, albeit at a significantly lower performance than the FicVer model. Given that pattern-based models naturally tend to lose information about previous sentences over sequences, this observation indicates that plot holes in stories are not local phenomena in general, and therefore require models capable of global synthesis in the case of longer stories. This conclusion accords with intuition as well and indicates we need a holistic approach.

Surprisingly, the worst-scoring models were the evidence-based models GET and DeClarE, which scored a $F1 = 0$ and an $F1$ comparable to that of guessing with random noise, respectively. One reason for the poor performance of

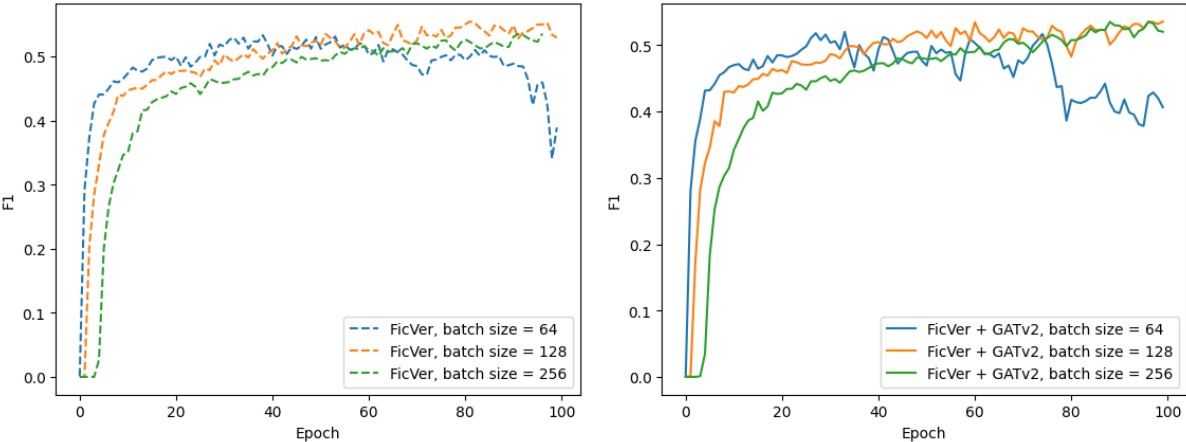

Figure 5: Analysis of the sensitivity of FicVer with and without the use of GNNs to batch size.

both of these models could be a result of the significant class imbalance in FicClaim, whereas the FEVER dataset that these models were originally developed for has a much more balanced class distribution (Thorne et al., 2018). Given that the rate of the positive class in the FicClaim dataset is on the order of 1 in 100 to 1 in 200, this imbalance could have heavily biased these models towards the negative class, thereby impeding their ability to learn. However, this reason does not fully explain why the DeClarE model performed similarly to noise, and further research is required to confirm this hypothesis. Another reason could be because the implementation of these models relies on the use of $O(n^2)$ memory with respect to the number of sentences in a story, and $O(n)$ memory with respect to the length of each sentence. This complexity meant that although we provided batch sizes of 100-200 to both of these models as described in Sec. 6.1.3, the highest batch size that was supported on a 32GB machine with respect to the actual FicClaim dataset was only 1. However, this technical drawback of the chosen models may indicate that they are not fit for application in fictional scenarios in the first place. Our overall conclusion is that existing approaches under-perform in the context of our problem and despite our results, further improvements via future research are expected.

## 7.2 FicVer Sensitivity

To offer a better insight about our model, we studied the sensitivity of models using GNNs with respect various hyperparameters. Most notably, we found that GNN-based models differed from those not using GNNs in their sensitivity to batch sizes. Our training results, shown in Fig. 5, indicate that models using GNNs to process KGs require batch sizes that are slightly larger than their non-GNN counterparts. Our experiments show that the non-GNN FicVer performed best with a batch size of 128, whereas GNN counterparts, such as the FicVer + GATv2 training shown in our results, performed better with a slightly larger batch size of 256.

## 7.3 The Problem Complexity of Multiple Plot Holes

In general, as is statistically expected and corroborated by the noise model results, finding single errors in fictional works tended to have a lower $F1$ score than with a higher number of errors with the weaker models. Surprisingly, however, for the stronger FicVer models, the inverse is true: as the number of errors increases, the difficulty in finding them also increased. We therefore find that as the performance of a model increases, its statistical correlation to noise breaks down and is instead replaced by what we posit is the "true" difficulty of solving the 2 and 5 plot hole scenarios. This observation makes sense, because as the number of injected errors increase, so does the probability that evidential opposing claims to that error are also replaced, therefore reducing (and in some cases eliminating) the evidence required to decide whether or not a sentence is a plot hole. For a rational model, this challenge makes the problem more difficult, whereas for a weaker randomly-guessing model, this property allows for a higher probability to guess at least one error correctly. Although we admit that this observation may be indicative of a shortcoming in the FicClaim dataset generation process itself, we nevertheless believe that settings with a higher rate of errors have strong practical consequence, as long-form media has a higher likelihood of having more than one error in it.

Therefore, although the number of errors in media can realistically be expected to occur at a rate much lower than even our 1 error per ∼200 sentences scenario, future research should focus on higher error rate configurations regardless, as they correspond to the hardest problems, and would likely enable the application of fictional plot hole detection and error correction models in the most impactful environments possible.

### 7.4 Current Feasibility for Industry Application

A significant driving force behind our research is the increasing acceptance of automated systems designed for detecting plot holes in entertainment industries. It's worth noting that, as of now, our work specifically pertains to text-based media and the prospect of adopting it highly depends on extending it to image and video inputs. However, the noteworthy achievement of FicVer in matching the scoring levels of human annotators suggests that our research has the potential for future applicability in industrial settings. We envision that the principles and methodologies developed in this research could contribute to the broader landscape of automated plot hole detection in diverse industrial applications. Since the scores of any of the other models besides FicVer significantly underperform relative to human annotators, plus the fact that we only test FicVer on short stories in this paper, further research will be required before adoption in real-world applications can occur. Furthermore, since human annotator scores themselves are relatively low, significant improvements could nevertheless be made to both FicVer and in research within this problem domain in general, both in terms of developing a better benchmark dataset and processing detection pipeline.

## 8 Discussion and Limitations

Despite the novelty of our work in being the first of itw own kind, we acknowledge the following limitations of our work, and envision future work for the purpose of mitigating and improving upon these limitations:

1. All of the documents in FicClaim are less than 300 sentences long, with the majority of them being under 100 sentences. The length of these documents is in stark contrast to the majority of real-world media, which are often hundreds of pages long. Thus, our work does not propose a framework or model that we claim could directly be applied in practical scenarios, but rather, ones that serve as a good introductory exploration into how to approach building them.Despite our best efforts, one major problem that we encountered with obtaining longer high-quality stories is the issue of copyright. Our existing budget also did not allow for generating 10s of thousands of long stories via LLMs such as ChatGPT. To help bridge this gap, we provide a script as a part of our public repository to scrape longer stories from the internet for future works, although we are unable to provide the data itself or run experiments on it as a part of this work.

2. Our current pipeline is only applicable to text-based media, whereas the majority of high-budget productions are either movies or TV shows. This limitation means that in the future, our work may need to be extended to support film formats in order to gain industry adoption.

3. Given that the KGs that we employ with FicVer are somewhat noisy, it is possible that future work focusing on denoising KGs could be applied to our method to yield stronger results. Since the focus of our work is on contributing a novel problem and dataset, as well as preliminary approaches to that dataset, we purposefully chose not to pursue any complex KG denoising strategies in this paper.

## 9 Conclusion

In this paper, we introduced the new problem domain of claim verification in fictional settings. We also proposed a framework for approaching this problem via generating a synthetic data to establish research in this area. We contribute FicClaim, a benchmark dataset created with the help of LLMs for learning how to detect plot holes in fictional settings, as well as FicVer, a new method to find logical inconsistencies in fictional works. We showed that FicVer outperforms existing pattern-based and evidence-based methods, thus suggesting that current methods of claim verification are not very extendable to the fictional claim verification problem domain. Given that the strongest model performances just barely match up with human-level performance for the toy scenario of short stories, which itself is quite low-scoring, we suggest that claim verification in fictional domains – especially for longer stories – may be a difficult problem in general. Future research includes accelerating the development of claim verification solutions for domains outside of social media and fake news detection, thereby advancing its adoption in high-impact entertainment applications.

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

# A  Appendix

## A.1  Alternative Sentence Negation Strategies

Besides using LLMs, a variety of alternate sentence negation strategies were attempted. The two other methods that we pursued before settling on LLMs were both forms of verb-level negation. Our first method simply added (or removed) a "not" before each verb, whereas our second method attempted to search for an antonym for the first verb:

1. Adding or removing "not" before every verb led to a large number of "not"s in the dataset, leading to poor sentence quality and biasing models. One major problem with this method was that while prepending every verb in a sentence with "not" was required to negate some sentences, in some cases, multiple "not"s cancelled each other out. For example, for simple sentences such as "I am tired." this method would correctly return "I am not tired." However, given the more complex sentence "I like to eat pizza", this method would incorrectly return "I not like to not eat pizza." Even ignoring the grammatical errors, the method introduces a double negative which leads to the sentence meaning the same thing.

2. In order to try to work around the self-cancelling effect of adding multiple "not"s to the same sentence, plus to improve the quality of the sentences, the second method that we tried was to attempt to replace only the first verb of a sentence with its antonym, and only add/remove "not" if no antonym was found. Formally, for a sentence $s$, let $v(s)$ be a function that returns the first verb of $s$. Also, let $A(w)$ be a function that returns a list of possible antonyms for some word $w$ ordered by relevance. For a verb $v$, we define $C(v)$ as the set of conjugations of $v$ (e.g., $C(\text{"be"}) = \{are, is, was...\}$). Our initial negation methods then replaced $v(s)$ with $C(v)$ as a way to perform semantic negation. Unfortunately, this method also led to poor sentence quality to the point where model performance was being affected.

Due to the difficulty in manually formulating a method to negate sentences, we eventually decided to employ LLMs to perform it instead. Despite the fact that LLM outputs cannot be predicted by us, this still led to much higher quality sentence outputs than before, based on the sample of outputs that we checked.

## A.2  FicClaim Sample

We provide a sample from the FicClaim dataset with comparisons to original sentences to showcase our method. In particular, we choose the following randomly-selected story:

1. *I wanted Keira, the head of the council, to come see.* $\rightarrow$ *I did not want Keira, the head of the council, to come see.* This sample displays our method's ability to perform simple negations in a similar fashion to our first verb negation methods.

2. *Or perhaps you did not meet your quota for Mana Reaping this Cycle?* $\rightarrow$ *And perhaps you did meet your quota for Mana Reaping this Cycle?* This sample shows that LLMs can change other words in a sentence to help keep the sentence natural-sounding in context (e.g. *Or* $\rightarrow$ *And*).

3. *I shook my head to clear it, "Mira needs a soother, please get her there as fast as possible," I said.* $\rightarrow$ *I didn't shake my head to clear it, "Mira doesn't need a soother".* This example shows two limitations of our method: first, that we cannot predict LLM outputs and therefore must sometimes use sub-optimal negations, and second, that we are forced to limit the response from OpenAI's API to a certain token length due to funding constraints.

## A.3  Real-World Applicability of FicClaim

Given that this work constitutes an introductory exploration into the problem domain of fictional claim verification, we specifically chose to pursue a set of assumptions during the FicClaim dataset generation process which would simplify and narrow the scope of the problem we are tackling. Therefore, while FicClaim is a good dataset for developing models for the purpose of exploring this domain, there are some departures from the practical reality of fictional claim verification, aside from the limitations discussed in Sec.8. Specifically, we assume that there is only one kind of plot

hole, i.e., continuity errors, and that the continuity errors are logically of a surface-level. We make this assumption for one main reason: this kind of plot hole is by far the simplest one to apply a synthetic data generation process for. This is both because sentence negation (a "surface-level" plot hole) is much easier to perform than deeper plot holes which may require multiple steps of logical inference in order to tease out; and also because it lends itself to a simple, sentence-level labelling scheme (i.e., the negated sentences), whereas deeper plot holes are likely to require more subtle labelling schemes. It is our expectation that more work will be necessary in the future to devise a suitable data generation and labelling scheme for more complicated continuity errors, as well as for other types of plot holes.

### A.4  Further Implementation Details

All models were trained on high-performance compute clusters using NVIDIA V100 GPUs. Some tuning was also performed on Microsoft Azure E2 and E4 virtual machines, depending on the amount of memory they needed to run without failures. Given that the data was originally sourced from Reddit, and consists mainly of the work of amateur writers – some of whom employ very long run-on sentences – we were forced to truncate sentences to the first 15 tokens in order to avoid significant slowdowns and out-of-memory (OOM) errors. We decided to specifically use the first 15 for two reasons: first, choosing 15 tokens was shown to cover a decent portion of the dataset ($\sim$80% of the total words), with higher numbers showing increasingly diminishing gains; and second, using higher values caused OOM errors for the evidence-based GET model, which requires $O(n)$ memory to run $w.r.t.$ sentence length, and $O(n^2)$ memory $w.r.t.$ the number of sentences per story. Given that the model already used $\sim$25GB-30GB of memory in the first place, pushing the token count past 15 would very likely cause OOMs on a 32GB machine after memory use from other processes is factored in. Since using a 64GB machine would have been prohibitively expensive for us, we decided to settle on the first 15 tokens for each sentence.

Given our constraints on funding, and because the OpenAI API charges by the number of tokens used, our specific implementation of the $InverseGPT()$ function requests that OpenAI limit the length of the response we receive from them to 20 tokens. As discussed above, however, this is more than enough to generate reasonable sentences that don't cut out too much information relative to the input sentence.

We also experimented with applying LLMs to the problem. However, our budgetary constraints disallowed us from training an LLM large enough to obtain any meaningful results, or alternatively from feeding our entire dataset to the OpenAI API. For that reason, we did not include any LLM results as a baseline, although working proof-of-concept code to generate such a baseline is available as a part of our code repository.

