# OpenReview forum: "FicClaim: A Framework for Claim Verification in Fictional Domains Using Synthetic Data Generation"
_TMLR — Rejected by TMLR_

### Review · Reviewer_HhvN · 2023-12-27

**Summary Of Contributions:**

The paper proposes claim verification in fictional domain, introduce a new synthetic dataset. The raw data (a set of story documents) is drawn from the web (r/writingprompts and r/stories) and then a varying number of sentences in each story document is manipulated to be “story holes”, meaning, the negation of the original sentence provided by GPT 3.5. The goal is to identify these story holes automatically (i.e., sentences that has been manipulated). They propose baseline methods, including one that uses graph neural network (based on graphs constructed from story preprocessing with standard classic NLP tools (named entity recognition, relation extraction), some existing methods such as TextCNN. Their proposed model (who GNN) outperform other approaches. I appreciate the idea of identifying sentences that are incoherent with overall story (this is related to some script learning literature that the paper is not referring too https://nlp.stanford.edu/pubs/narrative-chains08.pdf ). Having said that, I found the paper lacking severely in its clarity and experimental rigor, see my detailed comments.

**Audience:**

No

**Claims And Evidence:**

No

**Requested Changes:**

Please look at list of weaknesses/suggestions provided above.

**Strengths And Weaknesses:**

Strength:
* The paper proposes/addresses novel document-level task.

Weaknesses:
* The dataset is not properly described in the paper (e.g., dataset statistics are missing), and the quality of generated data is not evaluated. I think this is crucial missing piece, making it hard to recommend the paper in its current form.
* While the paper is motivated with misinformation, the proposed/presented work is very far from the complex real world misinformation. I highly doubt that progress/work on this dataset would translate at all to misinformation space. The proposed work seems to be more related to story coherence.
* The paper misses many crucial information, including the description of constructed KG and their qualities. For example, how many entities are there on average per each story? How many edges are there on average per edges? Do humans see this KGs agree they are of a good quality?
* The paper can be improved in terms of presentation and writing. They are often unclear and vague. For example, in the beginning of Section 3, what do you mean by “out-of-domain prediction relative to most existing training datasets”? Out-of-domain compared to what? What types of existing training datasets do you compare to? The space of prediction hasn’t been defined yet either (is it binary classification of being true or false?)? On page 4 “Story Encoding”, what do you mean by “high-fidelity” semantic sentence encoding? What is “continuity error” in Section 4 paragraph 4? In Section 8, it mentions that “somewhat noisy” — could you quantify? How did you reach the conclusion it is “somewhat” noisy? Many sentences are unclear, such as introducing acronyms (e.g., GET in related work “Claim verification” section) without spelling it out.

==
Minor comments:
The dataset sourcing part is a bit unclear, providing a link to Kaggle would be helpful (I assume Kaggle hosts Reddit data..?)

Hyperparameters for models are not provided (e.g., the size of transformer, lost, etc).

Eval metrics: F1 — is the altered sentence treated as the target label?

What is “one error dataset” that humans annotated? Did humans agree on this task? How large is this subset? If this is a subset, Table 1 should have subscript marking it clearly as the rows are not strictly comparable.

Figure 1 is a bit unclear, what is “Error”? What is 1-23? The labels should be clearly indexed.

I wonder “FicClaim” is the best name, maybe focus more on the aspect that it is story? My first impression of the name was that it would be about some financial documents or claims.

---

> ### Author Response · Authors · 2024-02-10
> **Response to the Reviewer**
>
> Thank you for your review and identifying the novelty of our work. First, addressing the weaknesses that you've pointed out sequentially:
>
> 1. We are happy to include any additional statistics if it would clarify our work. Could you please clarify which dataset statistics you would prefer to see (e.g., mean number of words per story, sentence length, etc)? As it stands, we already provide some details including the minimum number of words per document, the minimum number of upvotes for each post, the size of the dataset, and the precise synthetic generation method applied to it. Additionally, since the dataset will be public for future explorations, interested readers can extract any statistics that they found interesting.
>
> 2. We want to clarify that the point of our work, as described in the paper's introduction, is precisely the application of claim verification to plot hole detection for checking story coherence, rather for misinformation. Given that the purpose of the work is primarily concerned with application in the entertainment industry on *fictional* works, we do not find this to be a weakness. To be clear, the motivations which led to the development of our method are separate from the purpose of the work itself. In other words, literature on misinformation is extensive and we explore an area that has not been explored before.
>
> 3. Thank you for the note, we will update the work to provide these statistics about the KGs. Our team has viewed some of the KGs, and we have found that they reasonably reflect the story itself (understandably, we are unable to check every KG manually and we just checked instances of stories). Reinforcing that observation, our ablation study (Sec. 6.3, results in Table 1) indicates that the KGs themselves are very likely to have information which is important for understanding the story, given that the addition of the KG and GNNs improved our algorithm's performance. If it would help the clarity, these specific points can be added to the section describing our KG construction (in Sec. 5.1). Please let us know your thoughts.
>
> 4. We will make sure that some of these points (and especially the acronyms) are clarified. To address the specific points brought up:
>
> - Assuming we are understanding your question correctly, by "out-of-domain prediction relative to most existing training datasets", we simply mean that finding plot holes is out-of-domain as compared to most existing training datasets. As stated in the previous section (Sec. 2), we are specifically referring to knowledge repository datasets such as Wikipedia and Politifact (for example). If it improves the clarity, we can once again bring up these datasets as examples.
>
> - The space of prediction is defined in Section 3 (Problem Description). Specifically, that "we consider a solution to the problem to be the index...".
>
> - We use "high-fidelity" to describe the fact that the embeddings themselves are of a fairly large dimensionality of 384. This is not a technical term. If this description is causing confusion, we can remove it.
>
> - The term "continuity error" as used in Section 4 is defined in Problem Description section (Sec. 3).
>
> - The noisiness of the KGs is described in some detail in Sec. 5.1 under "Knowledge Graphs Construction". We are happy to provide a link back to this section if it helps to clarify what we are referring to. Unfortunately, it is difficult to quantify the actual noisiness or accuracy of the KGs since we do not have any ground truth/human annotator reference KGs with which to derive similarity metrics -- we only have the data from our ablation study, which indicates that the KGs are likely to contain useful information for our models.
>
> - Thank you for calling out our use of acronyms, we will fix this.
>
> Next, addressing each point in the "minor comments" section:
>
> - You are correct, we will provide a link to Kaggle.
>
> - Hyperparameters for all models are provided as a part of our GitHub repository (which is currently anonymized). We hope that it is sufficient to update the paper to mention this point, as opposed to writing another section/appendix to list the hyperparameters in the text.
>
> - Yes, the altered sentence(s) are treated as target labels. This is described in more detail in Sec. 3.
>
> - This is a good point, the subset of documents that were human-annotated were small relative to the full dataset, and thus only represent an approximation of human performance on that dataset. We will add a note to Table 1 clarifying this point as you have suggested.
>
> - We appreciate this feedback and will update Figure 1 to clarify what the indexes are. "P(Error | Sentence)" is a value described in Section 5.2.1.
>
> Finally, the narrative learning literature that you have linked is interesting and indeed could be relevant to our work. We will read it and update our related works as necessary.

---

### Review · Reviewer_78eu · 2024-01-05

**Summary Of Contributions:**

The main contributions of this paper are twofold. First, it introduced the new FicClaim dataset, which aims to assess machine learning models' ability to detect **plot holes / logical inconsistencies** on a given set of text. This problem differs from the standard fake news detection setup, as one cannot rely on the existence of external, commonly-established database of known, verified facts.

For example, in the Harry Potter world, magic exists, so the existence of magic is not a logical inconsistency in and of itself. In contrast, standard claims verification system would likely flag the existence of a magical world as a fake claim, which would be a false positive in this case. To construct the FicClaim dataset, the authors leveraged an LLM to **synthetically create** plot holes / logical inconsistencies by taking existing writing prompts, and negating some of the existing, human-written sentences, thereby introducing a logical inconsistency, with varying number of logical inconsistencies for each story (1, 2, or 5).

In the second part of the paper, the authors explored a few different approaches for solving this problem. The best-performing approach was to construct a set of knowledge graph triplets from the text using the StanfordNLP tool, and running the result through a graph neural network (GNN) to detect which of the sentences introduced the plot holes / logical inconsistencies. This approach was shown to outperform various baselines, such as pattern-based models (e.g. bidirectional LSTMs) that directly attempt to detect the sentences that introduced the plot holes from the text alone, without going through more structured representations & modelling (i.e. no knowledge graph construction & not using graph neural networks).

**Audience:**

Yes

**Broader Impact Concerns:**

No broader impact concerns come to mind.

**Claims And Evidence:**

Yes

**Requested Changes:**

**Critical**
1. More discussion around the points I raised regarding whether or not the dataset is a realistic representation of plot hole detection in the real world.

2. Resolving the questions / some unclear points I raised above.

**Strengths And Weaknesses:**

# Strengths

1. The use case of detecting plot holes & logical inconsistencies is a novel, interesting, and potentially very useful one, especially for writers, screenwriters, etc. Substantial progress here would make our LLM-based assistive agents even more useful for writing and content creation.

2. The paper is generally well-written and easy to follow.

3. The empirical setup is fairly rigorous, where the paper used statistical significance testing from 5 separate runs, and ran multiple different baselines and ablation studies.

# Weaknesses
While the use case is definitely an interesting & potentially high impact one, my main concern regarding the paper is the **artificial setup** of the dataset. In particular, the proposed dataset has several issues in my view:

1. First, the paper assumes that there is a **known** number of plot holes in the story. This does not seem to be a realistic assumption to me, and makes the problem substantially easier than the real-world use cases. In reality, there can be *any* number of plot holes in a story, from 0 (no plot holes) to many more than five. Therefore, determining the **number of plot holes** is very much an important part of the problem, which is not covered by this dataset.

2. Second, the paper assumes that there are **no plot holes in the original story** (e.g. from r/WritingPrompts and r/stories subreddits, page 3), and that **all plot holes are introduced by the LLM-generated synthetic data**. This assumption is not necessarily true: There can well be plot holes already in the original dataset, even before the LLM synthetic data process. After all, human-generated text is fairly likely to contain plot holes.

3. The introduced plot hole / logical inconsistency seems to only cover a small subset of possible logical inconsistencies. This is because the logical inconsistency is only taken by negating **individual sentences separately from one another**. Imagine that we have the following two sentences: "Bob is a frog", and "All frogs are green". It would be interesting to know if the model is able to tell whether "Bob is red" is a plot hole (because "Bob is a frog" and "All frogs are green" imply that "Bob is green", so the assertion that "Bob is red" would be a plot hole). These kinds of logical inconsistencies are crucially not covered by the dataset, which would have more limited & less interesting plot holes like "Bob is not a frog" or "Not all frogs are green", leaving out the models' ability to **detect plot holes that require a degree of reasoning**.

4. In reality, the script & stories would likely contain very long context, with an undetermined number of plot holes. Some of these plot holes would also require a degree of reasoning to detect (i.e. it is much more difficult to detect than the plot holes that simply take the negation of existing sentences). All in all, I am not sure if the dataset is a good and realistic benchmark for evaluating progress in this direction, and that progress in this dataset would necessarily transfer to better plot hole detection in real-world scenarios.

# Questions & Other Weaknesses
Beyond how realistic the dataset is, some other aspects of the paper are also still unclear to me. In particular:

1. The knowledge graph triplet extraction is based on the StanfordNLP toolkit, which used a pipeline system to do so. The problem with this approach is that the pipeline system can make a lot of errors, which may propagate to the later stages of the approach & make the approach less accurate. Do you have a sense of how much of the error is due to incorrect knowledge graph triplets?

2. How exactly is the loss computed? Is the probability of p(error | sentence) calculated independently for each sentence? Or do you take a softmax over all sentences' error probabilities, and only take the top {1, 5} sentences with the highest probabilities of being plot holes? It would be nice to spell out the loss function more clearly.

3. For the equations on page 6, I would recommend using a more standard notation: Bold uppercase letters for matrices, bold lowercase letters for vectors, and non-bold letters for scalars, etc.

4. In Table 1, how exactly does the "FicVer" model work (as opposed to "FicVer + GCN" and "FicVer + GATv2", which I assume used graph neural networks)? Does the "FicVer" model not use graph neural networks? But does it still use the knowledge graphs, or does it work purely based on textual / non-structured representation? If it's the latter, how does it differ from the other baselines?

5. For Figure 5, did you vary the learning rate for different batch sizes? Usually learning with larger batch sizes would require a higher learning rate (due to there being fewer updates overall).

# Typo
On page 2, "depthful" is not really a word. Maybe "A more depthful semantic knowledge" can be changed to "A deeper semantic knowledge"?

---

> ### Author Response · Authors · 2024-02-10
> **Response to the Reviewer**
>
> Thank you for the comprehensive review of our work. We appreciate that you have identified our work to be novel and our empirical explorations comprehensive. We have tried to address the concerns you have raised. In the paper, our changes are marked in blue. We have responded to your questions below:
>
> 1. It is true that errors from StanfordNLP could ripple downstream and affect our models. However, we find it hard to quantify that error – presumably, relative to some "label" graph -- because we don't have any reference graphs for any of our stories to make comparisons. If we had human annotator graphs (for example) for comparison, then it is possible that we could derive some similarity metrics and possibly quantify that error, but we do not have the resources to procure them. We only know, as indicated by our ablation study, that the graphs are likely to contain some amount of valuable information, given that the strongest models were using them. Do you think that including this mention would clarify our method (e.g., in Sec 5.1) and address this concern? We also would like you to consider that when a work is a first of its own kind, prior resources are scarce and coming up with a fully realistic solution is not straightforward. Many similar works with this aspect have a proof-of-concept nature as does our work.
>
> 2. To calculate the loss, we take a softmax over the sentences. We then use a pr-threshold of 0.3 to decide whether or not a sentence is a "plot hole" or not. This point will be clarified in the paper.
>
> 3. Thank you for the feedback, we update these equations.
>
> 4. You are correct that the model labeled "FicVer" doesn't use any GNNs. The main difference from other baselines is that it uses sentence-level embeddings. The reason we provide this result is to quantify how much the GNNs are actually contributing to the final performance.
>
> 5. We used the same learning rate (1e-5) for all experiments in Figure 5, as it was experimentally the strongest. We used grid search to find the best learning rate values, jumping by factors of 10 (i.e. 1e-3, 1e-4, etc). To your concern, we can run a finer search for Figure 5 to confirm our results (e.g., by factors of 2 since the batch sizes differ by factors of 2).
>
>
> Second, to address the weaknesses that you have pointed out:
>
> 1. You are correct. As this work is the first in this area, we have intentionally simplified the problem. We will mention this in the paper, and further point out that our dataset could, in fact, still be used for the more complicated version of the problem via mixing the {1, 2, 5}-plot hole documents (as well as the original documents, for 0 plot holes).
>
> 2. It is true that there are likely to be some number of plot holes already sprinkled throughout the dataset -- however, we take steps to limit these by constraining the original stories to those that have at least 1,000 upvotes as a measure of quality. Although this will not eliminate them, we hope that this would reduce the number of pre-existing plot holes and thereby increase the accuracy of our synthetic labels (under the assumption that stories with plot holes are less likely to receive upvotes). In addition to clarifying this point in the paper, if it would strengthen our work, we could also mention an alternative approach to reducing the number of plot holes in the original dataset as future work by generating *all* original stories using LLMs.
>
> 3. This is certainly a very interesting point. We have intentionally chosen to pursue this specific (and admittedly small subset of) plot hole because it simplifies our exploration of this new problem domain. To provide an example of the complexity that we are trying to avoid (for now), let us consider your example (A: "Bob is a frog", B: "All frogs are green", C: "Bob is red"). Since we don't actually know which claims are the truth, it is actually possible that the plot hole could be any of the three claims (i.e. A+B => not C; A+C => not B; B+C => not A). Of course, it is probable that the additional context provided by a full story would shift the likelihood of the plot hole to one set of sentences over another set, but even so, the method of creating a dataset for this kind of plot hole, as well as accurately choosing a synthetic labeling strategy which would hopefully reflect this subtlety, are complex tasks which we did not want to tackle in this work. However, we are willing to include this discussion in the paper in order to clarify the kind of future work that would be necessary for practical application -- would that help address this concern?
>
> 4. We hope that this concern is a summary of 1-3 and is hopefully alleviated by our previous responses
>
> Finally, thank you for pointing out the typo.

---

> > ### Comment · Reviewer_78eu · 2024-02-27
> > **Thank You for the Response**
> >
> > Thank you for the response. I understand that resource limitations are an impediment to addressing some of the points (in particular, for making the dataset more realistic and reflective of real-world use cases, such as exploring more sophisticated types of plot holes, etc., in addition to quantifying how accurate the StanfordNLP graph construction pipeline is).
> >
> > I think some of the discussion points we have here would benefit from inclusion in the paper. This includes discussions around: (i) whether or not the constructed knowledge graph is likely to be accurate (the authors made a good point that including this knowledge graph helps, so it likely contains some useful information); (ii) the selection criterion to only include highly-voted stories, under the assumption that these are less likely to contain plot holes in the first place; (iii) extension to long context stories & how to make the dataset more realistic in the future; clarification around the loss and what the "FicVer" baseline is; etc.
> >
> > I see that other reviewers also raise concerns around whether or not the dataset is reflective of real-world use cases, so in my view, a comprehensive discussion around this is the most important change to make.

---

> > > ### Author Response · Authors · 2024-02-28
> > >
> > > As suggested, we've added these discussions to our work. In particular, we have also added a short discussion about the real-world usability of the dataset in the appendix (including relevant pointers to that section within the main body of the work). Thank you once again for your feedback!

---

### Review · Reviewer_6TPw · 2024-01-31

**Summary Of Contributions:**

* Contributes a new dataset about fact verification. Instead of focusing on traditional fact verification tasks focusing on news, daily life knowledge, etc. This dataset specifically focuses on the "plot holes" in fictional stories, which tend to be hard for previous methods to tackle due to their context-dependent properties.

* Contributes a new graph-based algorithm for factual verification which only utilizes a relatively small portion of training data and gets a performance better than the baselines reported in the paper.

**Audience:**

Yes

**Broader Impact Concerns:**

There's no concerns on the ethical implications of the work.

**Claims And Evidence:**

Yes

**Requested Changes:**

* A more diverse dataset containing longer stories that reflect the inherent challenge of plot hole detection in fiction.

* Baselines of utilizing LLMs to do the task.

**Strengths And Weaknesses:**

## Strengths

* Introduces an interesting task and dataset that could potentially benefit the entertainment industry for better script writing and plot design. Such tasks are often overlooked since most fact-verification works solely focus on real-life fact-verification.

* Designs initial algorithms that specifically target this problem which could potentially inspire future methods improving on this problem.

## Weaknesses

* The way (i.e., negating some sentences in the text) how the dataset is made seems to be targeting the surface-level plot holes that could potentially exist in the fictional stories. Meanwhile, such negation may create artifacts that make the problem less challenging. I am not sure if algorithms that are doing really well on this dataset actually imply real-world utility for the entertainment industry.

* As illustrated in the discussion section of the paper, the collected dataset has a majority of stories under 100 sentences, which does not reflect the real challenge of detecting plot holes in the created fictional world. This also raises another question about the baselines the paper is comparing: Since the majority of stories can fit in well in a current large language model, why there's no experimentation and comparison about using LLMs to do this task?

---

> ### Author Response · Authors · 2024-02-10
> **Response to the Reviewer**
>
> Thank you for highlighting the strength of our work and clarifying its distinction from prior works on this domain. We have tried to address the concerns you have raised. In the paper, our changes are marked in blue. Specific to your concerns:
>
> 1. As you have mentioned, our work is the first work in this problem domain and there are not prior resources that we could benefit from. For this reason, we have purposefully chosen to scope our work to the specific type of plot hole on shorter stories as a baseline for future efforts. Broadening our scope would require access to human labor resources because tools such as ChatGPT are not strong on lengthy texts. Unfortunately, our resources are limited on this end. Additionally, as we have discussed in our limitations section, it is difficult for us to obtain longer stories due to issues with copyright and the terms of service of most sites with respect to web scraping. Furthermore, even if we did obtain a dataset of longer stories, we do not have the computational resources to train all of the models on such a dataset. We are affiliated with an academic institution and the allocation for this project is limited. That being said, we actually do have a script that we prepared for the purpose of scraping longer stories (>10k words, >1k upvotes) from a fanfiction website that could be used to obtain a dataset of much longer stories for future research (given the permission of the respective site administrators at some point in the future). This script will be uploaded to the repository (which is currently anonymized), and instructions for its use will be clearly posted in the readme. We hope that our explanations address your concern and you take into consideration that there are limits for the first works in all domains when prior resources are lacking.
>
> 2. Using LLMs to tackle this problem is an interesting idea. Although we considered this idea before submitting our work, there are two main reasons why we chose not to go down the path of using an LLM as a baseline. First, we do not have the budget to use the latest LLMs such as ChatGPT, and would thus have to resort to running a weaker LLM locally to obtain these results (e.g., a pretrained LLaMA-7b) which likely will not lead to competitive results. Second, even if we chose to try to obtain results from a pretrained LLaMA-7b or other similar models, there is still the question of prompt engineering. For example, it is possible that performance would be different when prompting with "Give us the indices of sentences that are plot holes: {story}" vs "Which sentence indices have plot holes?: {story}" vs "Which sentences have have plot holes? 0: {sentence0}, 1: {sentence1}, ...", etc. Thus, we would have to arbitrarily choose some set of prompts to test which may not necessarily lead to the optimal performance by an LLM. Although this problem also exists for our dataset generation method, it was easy for us to quickly verify whether or not a prompt was reasonable since the task of negating a single sentence is (relatively) much simpler. In other words, although it is possible to provide an LLM baseline, we are not confident that it would be a reasonable representation of LLM performance on the dataset. Regardless, we can attempt to provide such a baseline by the end of the review period if possible. Given our explanations, do you think that doing so would significantly strengthen our work?

---

> > ### Comment · Reviewer_6TPw · 2024-02-26
> >
> > Yes, providing such baselines would be helpful, and explaining your budget constraint in the paper would also be good.

---

> > > ### Author Response · Authors · 2024-02-28
> > >
> > > As you have suggested, we have added comments on our constraints in our appendix. Although we tried providing a complete LLM baseline, we ran into memory issues running one. On the bright side, we now have working, end-to-end proof-of-concept code in our repository, which could, in theory, generate such a baseline given enough resources. We have additionally also mentioned the availability of this code for an LLM baseline in the paper.
> > >
> > > To provide a little bit of additional insight into our development process, we found that we were unable to train any LLM baselines, and the only pre-trained LLM baseline that we were actually able to run in inference mode (~31M parameters) ended up being too small to understand the stories and consequently ended up spitting out nonsense every single time (e.g., a repetition of "I'm sorry. I'm sorry. I'm sorry", etc).
> > >
> > > Finally, we believe that this addition has improved the strength of our work and want to thank you once again for your suggestions!

---

### Author Response · Authors · 2024-02-10
**General Note to the Reviewers**

Dear Reviewers,

We thank all reviewers for their time and effort. We are aware reviewers are doing this service voluntarily and appreciate their feedback that helped improving our work. Judging from the quality of the feedback we have received, we are glad that ALL reviewers have read our work carefully. We are glad that ALL the reviewers have found our work to be novel. There are of course raised concerns and weaknesses. However, we think that all these concerns can be addressed through continual engagement with the reviewer. According to the AE, we will have about 18 more days to continue our discussion. We are hopeful to address all the concerns during this period.

We have provided responses in each individual review. Additionally, we have updated the manuscript and the changes are marked blue.

Thank you,

Our team

---

### Decision · Action_Editor_3xVE · 2024-03-04

**Recommendation:** Reject

**Comment:**

Reviewers acknowledged the detailed authors' response and appreciated the recognition of resource limitations as a key impediment to creating a more realistic dataset and conducting thorough manual quality assurance.

However, there are real risks associated with the community proposing models that perform well on this dataset in the future and assuming significant progress in this space. This progress may not necessarily translate well to real-life plot hole detection use cases due to the artificial setup of the dataset, which may not measure progress on the aspects that are most relevant.

As a result, all three reviewers proposed rejection.

**Audience:**

The findings could stimulate discussions on plot hole detection, with the aim of encouraging more researchers to delve into this area and develop improved datasets in the future.

**Claims And Evidence:**

The manuscript proposes a new dataset focusing on the plot holes (logical inconsistencies) in fictional stories. This problem differs from the standard fake news detection setup. Unlike fake news detection, where one can rely on the existence of an external, commonly-established database of known, verified facts, in detecting plot holes, such databases are unavailable. The manuscript also explores several approaches for detecting plot holes.

While the proposed dataset is interesting and novel, there are major concerns: Firstly, the setup is rather narrow, focusing on short stories, whereas real movie scripts can span many pages. Secondly, the type of plot holes identified seems artificial and likely corresponds to those that humans can already easily detect by skimming through the script. Finally, the quality assurance process of the dataset appears insufficient. For instance, the original dataset may already contain plot holes, and errors in the graph extraction pipeline may propagate further inaccuracies.